# PLUG-IN INVERSION: MODEL-AGNOSTIC INVERSION FOR VISION WITH DATA AUGMENTATIONS

## ABSTRACT

Existing techniques for model inversion typically rely on hard-to-tune regularizers, such as total variation or feature regularization, which must be individually calibrated for each network in order to produce adequate images. In this work, we introduce Plug-In Inversion, which relies on a simple set of augmentations and does not require excessive hyper-parameter tuning. Under our proposed augmentation-based scheme, the same set of augmentation hyper-parameters can be used for inverting a wide range of image classification models, regardless of input dimensions or the architecture. We illustrate the practicality of our approach by inverting Vision Transformers (ViTs) and Multi-Layer Perceptrons (MLPs) trained on the ImageNet dataset, tasks which to the best of our knowledge have not been successfully accomplished by any previous works.

## 1 INTRODUCTION

Model inversion is an important tool for visualizing and interpreting behaviors inside neural architectures, understanding what models have learned, and explaining model behaviors. In general, model inversion seeks inputs that either activate a feature in the network (*feature visualization*) or yield a high output response for a particular class (*class inversion*) (Olah et al., 2017). Model inversion and visualization has been a cornerstone of conceptual studies that reveal how networks decompose images into semantic information (Zeiler & Fergus, 2014; Dosovitskiy & Brox, 2016). Over time, inversion methods have shifted from solving conceptual problems to solving practical ones. Saliency maps, for example, are image-specific model visualizations that reveal the inputs that most strong influence a model's decisions (Simonyan et al., 2014).

Recent advances in network architecture have posed major challenges for existing model inversion schemes. Convolutional Neural Networks (CNN) have long been the de-facto approach for computer vision tasks, and are the focus of nearly all research in the model inversion field. Recently, other architectures have emerged that achieve results competitive with CNNs. These include Vision Transformers (ViTs; Dosovitskiy et al., 2021), which are based on self-attention layers, and MLP-Mixer (Tolstikhin et al., 2021) and ResMLP (Touvron et al., 2021a), which are based on Multi Layer Perceptron layers. Unfortunately, most existing model inversion methods either cannot be applied to these architectures, or are known to fail. For example, the feature regularizer used in DeepInversion (Yin et al., 2020) cannot be applied to ViTs or MLP-based models because they do not include Batch Normalization layers (Ioffe & Szegedy, 2015).

In this work, we focus on class inversion, the goal of which is to find interpretable images that maximize the score a classification model assigns to a chosen label without knowledge about the model's training data. Class inversion has been used for a variety of tasks including model interpretation (Mordvintsev et al., 2015), image synthesis (Santurkar et al., 2019), and data-free knowledge transfer (Yin et al., 2020). However, current inversion methods have several key drawbacks. The quality of generated images is often highly sensitive to the weights assigned to regularization terms, so these hyper-parameters need to be carefully calibrated for each individual network. In addition, methods requiring batch norm parameters are not applicable to emerging architectures.

To overcome these limitations, we present *Plug-In Inversion* (*PII*), an augmentation-based approach to class inversion. *PII* does not require any explicit regularization, which eliminates the need to tune regularizer-specific hyper-parameters for each model or image instance. We show that *PII* is able to

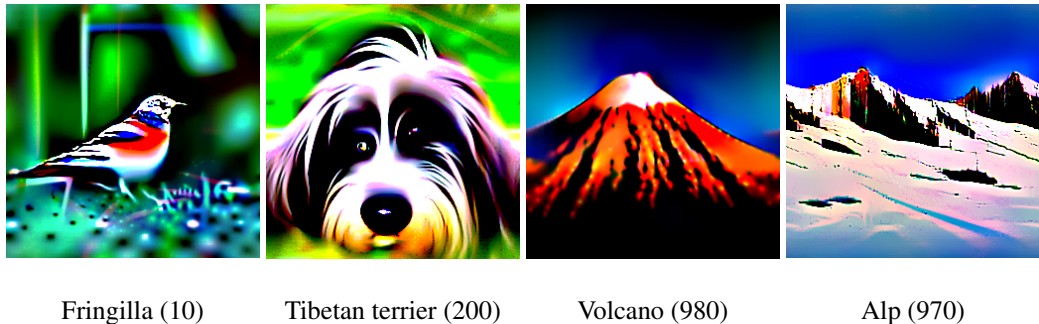

| Fringilla (10) | Tibetan terrier (200) | Volcano (980) | Alp (970) |

Figure 1: Inverted images from a robust-ResNet50 model trained on ImageNet-1k.

invert CNNs, ViTs, and MLP networks using the same architecture-agnostic method, and with the same architecture-agnostic hyper-parameters.

### 1.1 CONTRIBUTIONS

We summarize our contributions as follows:

- We provide a detailed analysis of various augmentations and how they affect the quality of images produced via class inversion.
- We introduce *Plug-In Inversion* (*PII*), a new class inversion technique based on these augmentations, and compare it to existing techniques.
- We apply *PII* to dozens of different pre-trained models of varying architecture, justifying our claim that it can be 'plugged in' to most networks without modification.
- In particular, we show that *PII* succeeds in inverting ViTs and large MLP-based architectures, which to our knowledge has not previously been accomplished.
- Finally, we explore the potential for combining *PII* with prior methods.

In section 2, we review existing techniques for class inversion and outline the types of architectures we consider. In section 3, we explore individually the different augmentations that we use, and then describe the full *PII* algorithm. The remainder is devoted to results and analysis.

## 2 BACKGROUND

### 2.1 CLASS INVERSION

In the basic procedure for class inversion, we begin with a pre-trained model $f$ and chosen target class $y$. We randomly initialize (and optionally pre-process) an image $\mathbf{x}$ in the input space of $f$. We then perform gradient descent to solve the following optimization problem for a chosen objective function $\mathcal{L}$,

$$\hat{x} = \arg\min_{\mathbf{x}} \mathcal{L}(f(\mathbf{x}), y),$$

and the result is class image $\hat{x}$. For very shallow networks and small datasets, letting $\mathcal{L}$ be cross-entropy or even the negative confidence assigned to the true class can produce recognizable images with minimal pre-processing (Fredrikson et al., 2015). Modern deep neural networks, however, cannot be inverted as easily.

### 2.2 REGULARIZATION

Most prior work on class inversion for deep networks has focused on carefully designing the objective function to produce quality images. This entails combining a divergence term (e.g. cross-entropy) with one or more regularization terms (*image priors*) meant to guide the optimization towards an image with 'natural' characteristics. *DeepDream* (Mordvintsev et al., 2015), following

work on feature inversion (Mahendran & Vedaldi, 2015), uses two such terms: $\mathcal{R}_{\ell_2}(\mathbf{x}) = \|\mathbf{x}\|_2^2$, which penalizes the magnitude of the image vector, and total variation, defined as[1]

$$\mathcal{R}_{TV}(\mathbf{x}) = \left(\sum_{i,j}(x_{i+1,j} - x_{i,j})^2\right)^{\frac{1}{2}} + \left(\sum_{i,j}(x_{i,j+1} - x_{i,j})^2\right)^{\frac{1}{2}}$$
$$+ \left(\sum_{i,j}(x_{i+1,j+1} - x_{i,j})^2\right)^{\frac{1}{2}} + \left(\sum_{i,j}(x_{i+1,j} - x_{i,j+1})^2\right)^{\frac{1}{2}},$$

which penalizes sharp changes over small distances. *DeepInversion* (Yin et al., 2020) uses both of these regularizers, along with the feature regularizer

$$\mathcal{R}_{feat}(\mathbf{x}) = \sum_{k}\left(\|\mu_k(\mathbf{x}) - \hat{\mu}_k\|_2 + \|\sigma_k^2(\mathbf{x}) - \hat{\sigma}_k^2\|_2\right),$$

where $\mu_k, \sigma_k^2$ are the batch mean and variance of the features output by the $k$-th convolutional layer, and $\hat{\mu}_k, \hat{\sigma}_k^2$ are corresponding Batch Normalization statistics stored in the model (Ioffe & Szegedy, 2015). Naturally, this method is only applicable to models that use Batch Normalization, which leaves out ViTs, MLPs, and even some CNNs. Furthermore, the optimal weights for each regularizer in the objective function vary wildly depending on architecture and training set, which presents a barrier to easily applying such methods to a wide array of networks.

## 2.3 ARCHITECTURES FOR VISION

We now present a brief overview of the three basic types of vision architectures that we will consider.

**Convolutional Neural Networks** (CNNs) have long been the standard in deep learning for computer vision (LeCun et al., 1989; Krizhevsky et al., 2012). Convolutional layers encourage a model to learn properties desirable for vision tasks, such as translation invariance. Numerous CNN models exist, mainly differing in the number, size, and arrangement of convolutional blocks and whether they include residual connections, Batch Normalization, or other modifications (He et al., 2016; Zagoruyko & Komodakis, 2016; Simonyan & Zisserman, 2014).

Dosovitskiy et al. (2021) recently introduced **Vision Transformers** (ViTs), adapting the transformer architectures commonly used in Natural Language Processing (Vaswani et al., 2017). These models break input images into patches, combine them with positional embeddings, and use these as input tokens to self-attention modules. Others have proposed variants which require less training data (Touvron et al., 2021c), have convolutional inductive biases (d'Ascoli et al., 2021), or make other modifications to the attention modules (Chu et al., 2021; Liu et al., 2021b; Xu et al., 2021).

Subsequently, a number of authors have proposed vision models which are based solely on **Multi-Layer Perceptrons** (MLPs), using insights from ViTs (Tolstikhin et al., 2021; Touvron et al., 2021a; Liu et al., 2021a). Generally, these models use patch embeddings similar to ViTs and alternate channel-wise and patch-wise linear embeddings, along with non-linearities and normalization.

We emphasize that as the latter two architecture types are recent developments, our work is the first to study them in the context of model inversion.

## 3 PLUG-IN INVERSION

Prior work on class inversion uses augmentations like jitter, which randomly shifts an image horizontally and vertically, and horizontal flips to improve the quality of inverted images (Mordvintsev et al., 2015; Yin et al., 2020). The hypothesis behind their use is that different views of the same image should result in similar scores for the target class. These augmentations are applied to the input before feeding it to the network, and different augmentations are used for each gradient step

---

[1]This is the formulation used by (Yin et al., 2020); others are also common, such as the simpler version in (Mahendran & Vedaldi, 2015).

used to reconstruct $x$. In this section, we explore additional augmentations that benefit inversion before describing how we combine them to form the *PII* algorithm.

As robust models are typically easier to invert than naturally trained models (Santurkar et al., 2019; Mejia et al., 2019), we use a robust ResNet-50 (He et al., 2016) model trained on the ImageNet (Deng et al., 2009) dataset throughout this section as a toy example to examine how different augmentations impact inversion. Note, we perform the demonstrations in this section under slightly different conditions and with different models than those ultimately used for *PII* in order to highlight the effects of the augmentations as clearly as possible. The reader may find thorough experimental details in the appendix, section D.

## 3.1 RESTRICTING SEARCH SPACE

In this section, we consider two augmentations to improve the spatial qualities of inverted images: *Centering* and *Zoom*. These are designed based on our hypothesis that restricting the input optimization space encourages better placement of recognizable features. Both methods start with small input patches, and each gradually increases this space in different ways to reach the intended input size. In doing so, they force the inversion algorithm to place important semantic content in the center of the image.

**Centering** Let $x$ be the input image being optimized. At first, we only optimize a patch at the center of $x$. After a fixed number of iterations, we increase the patch size outward by padding with random noise, repeating this until the patch reaches the full input size. Figure 2 shows the state of the image prior at each stage of this process, as well as an image produced without centering. Without centering, the shift invariance of the networks allows most semantic content to scatter to the image edges. With centering, results remain coherent.

| | | | | w/ Centering | | | | | w/o Centering |
| Init | Step 1 | Step 2 | Step 3 | Step 4 | Step 5 | Step 6 | Step 7 | Final | Final |

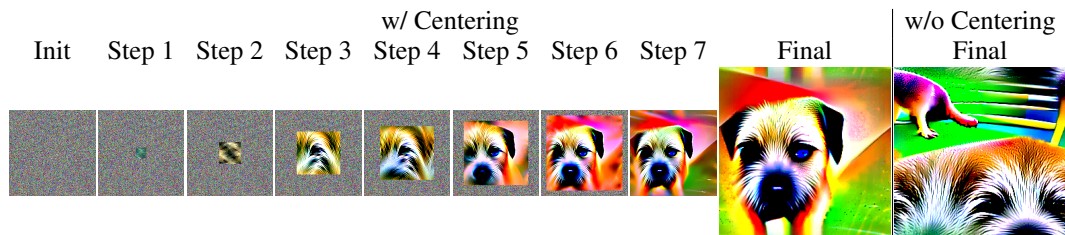

Figure 2: An image at different stages of optimization with centering (left), and an image inverted without centering (right), for the Border Terrier class of a robust ResNet-50.

**Zoom** For zoom, we begin with an image $x$ of lower resolution than the desired result. In each step, we optimize this image for a fixed number of iterations and then up-sample the result, repeating until we reach the full resolution. Figure 3 shows the state of an image at each step of the zoom procedure, along with an image produced without zoom. The latter image splits the object of interest at its edges. By contrast, zoom appears to find a meaningful structure for the image in the early steps and refines details like texture as the resolution increases.

We note that zoom is not an entirely novel idea in inversion. Yin et al. (2020) use a similar technique as 'warm-up' for better performance and speed-up. However, we observe that continuing zoom throughout optimization contributes to the overall success of *PII*.

**Zoom + Centering** Unsurprisingly, we have found that applying zoom and centering simultaneously yields even better results than applying either individually, since each one provides a different benefit. Centering places detailed and important features (e.g. the dog's eye in Figure 2) near the center and builds the rest of the image around the existing patch. Zoom helps enforce a sound large-scale structure for the image and fills in details later.

The combined Zoom and Centering process proceeds in 'stages', each at a higher resolution than the last. Each stage begins with an image patch generated by the previous stage, which approximately minimizes the inversion loss. The patch is then up-sampled to a resolution halfway between the

w/ Zoom

w/o Zoom

Init  Step 1  Step 2  Step 3  Step 4  Step 5  Step 6  Step 7  Final  Final

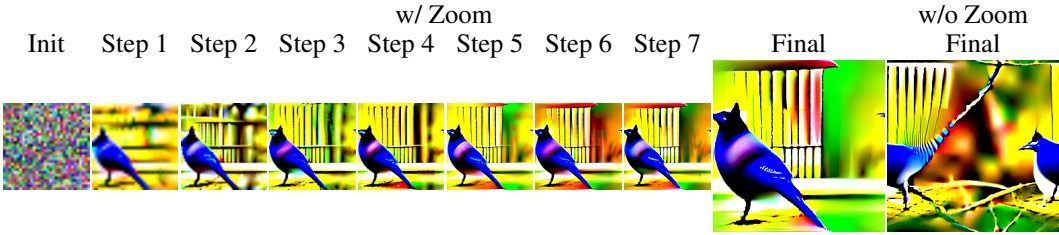

Figure 3: An image during different stages of optimization with zoom (left), and an image inverted without zoom (right), for the Jay class of a robust ResNet-50.

previous stage and current stage resolution, allowing it to fill the center of the image, leaving a border which is padded with random noise. Then next round of optimization then begins starting from this newly processed image.

## 3.2 COLORSHIFT AUGMENTATION

The colors of the illustrative images we have shown so far are notably different from what one might expect in a natural image. This is due to *ColorShift*, a new augmentation that we now present.

ColorShift is an adjustment of an image's colors by a random mean and variance in each channel. This can be formulated as follows:

$$\text{ColorShift}(\mathbf{x}) = \sigma \mathbf{x} - \mu,$$

where $\mu$ and $\sigma$ are $C$-dimensional[2] vectors drawn from $\mathcal{U}(-\alpha, \alpha)$ and $e^{\mathcal{U}(-\beta, \beta)}$, respectively, and are repeatedly redrawn after a fixed number of iterations. We use $\alpha = \beta = 1.0$ in all demonstrations unless otherwise noted. At first glance, this deliberate shift away from the distribution of natural images seems counterproductive to the goal of producing a recognizable image. However, our results show that using ColorShift noticeably increases the amount of visual information captured by inverted images and also obviates the need for hard-to-tune regularizers to stabilize optimization.

We visualize the stabilizing effect of ColorShift in Figure 4. In this experiment, we invert the model by minimizing the sum of a cross entropy and a total-variation (TV) penalty. Without ColorShift, the quality of images is highly dependent on the weight $\lambda_{TV}$ of the TV regularizer; smaller values produce noisy images, while larger values produce blurry ones. Inversion with ColorShift, on the other hand, is insensitive to this value and in fact succeeds when omitting the regularizer altogether.

$log(\lambda_{tv})$ :    −9      −8      −7      −6      −5      −4

w/ CS

w/o CS

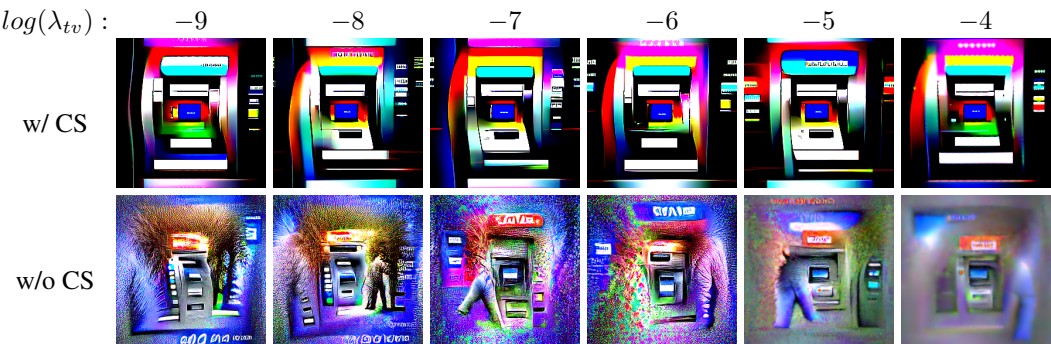

Figure 4: Inversions of the robust ResNet-50 ATM class, with and without ColorShift and with varying TV regularization strength. The inversion process with ColorShift is robust to changes in the $\lambda_{tv}$ hyper-parameter, while without it, $\lambda_{tv}$ seems to present a trade-off between noise and blur.

---

[2] $C$ being the number of channels

$e = 1$     $e = 2$     $e = 4$     $e = 8$     $e = 16$     $e = 32$     $e = 64$

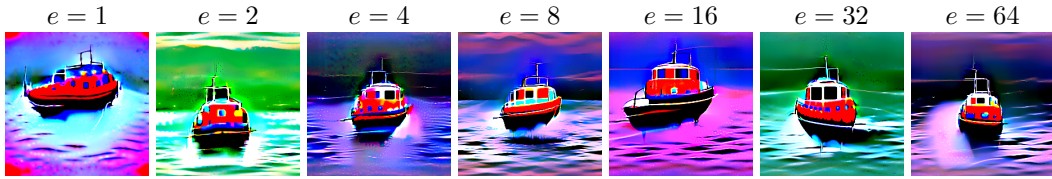

Figure 5: Effect of ensemble size in the quality of inverted images for the Tugboat class of a robust ResNet-50.

Other preliminary experiments show that ColorShift similarly removes the need for $\ell_2$ or feature regularization, as our main results for *PII* will show. We conjecture that by forcing unnatural colors into an image, ColorShift requires the optimization to find a solution which contains meaningful semantic information, rather than photo-realistic colors, in order to achieve a high class score. Alternatively, as seen in Figure 9, images optimized with an image prior may achieve high scores despite a lack of semantic information merely by finding sufficiently natural colors and textures.

### 3.3 ENSEMBLING

Ensembling is an established tool often used in dataset security (Souri et al., 2021) to enhanced inference (Opitz & Maclin, 1999). Similarly, we find that optimizing an ensemble composed of different ColorShifts of the same image simultaneously improves the performance of inversion methods. To this end, we minimize the average of cross-entropy losses $\mathcal{L}(f(\mathbf{x}_i), y)$, where the $\mathbf{x}_i$ are different ColorShifts of the image at the current step of optimization. Figure 5 shows the result of applying ensembling alongside ColorShift. We observe that larger ensembles appear to give slight improvements, but even ensembles of size 1 or two produce satisfactory results. This is important for models like ViTs, where available GPU memory constrains the possible size of this ensemble; in general, we use the largest ensemble size (up to a maximum of $e = 32$) that our hardware permits for a particular model. More results on the effect of ensemble size can be found in Figure 14. We show the effect of ensembling using other well-known augmentations and compare them to ColorShift in Appendix Section A.5. We empirically show that ColorShift is the strongest among augmentations we tried for model inversion.

### 3.4 THE PLUG-IN INVERSION METHOD

We combine the jitter, ensembling, ColorShift, centering, and zoom techniques, and name the result Plug-In Inversion, which references the ability to 'plug in' any differentiable model, including ViTs and MLPs, using a single fixed set of hyper-parameters. In the next section, we detail the experimental method that we used to find these hyper-parameters, after which we present our main results.

## 4 EXPERIMENTAL SETUP

In order to tune hyper-parameters of *PII* for use on naturally-trained models, we use the torchvision (Paszke et al., 2019) ImageNet-trained ResNet-50 model. We apply centering + zoom simultaneously in 7 'stages.' During each stage, we optimize the selected patch for 400 iterations, applying random jitter and ColorShift at each step. We use the Adam (Kingma & Ba, 2014) optimizer with momentum $\beta_m = (0.5, 0.99)$, initial learning rate $lr = 0.01$, and cosine-decay. At the beginning of every stage, the learning rate and optimizer are re-initialized. We use $\alpha = \beta = 1.0$ for the ColorShift parameters, and an ensemble size of $e = 32$. We ablate these choices in Figure 6 to validate our selected settings. Further ablation studies can be found in the appendix in figures 10, 13, and 14.

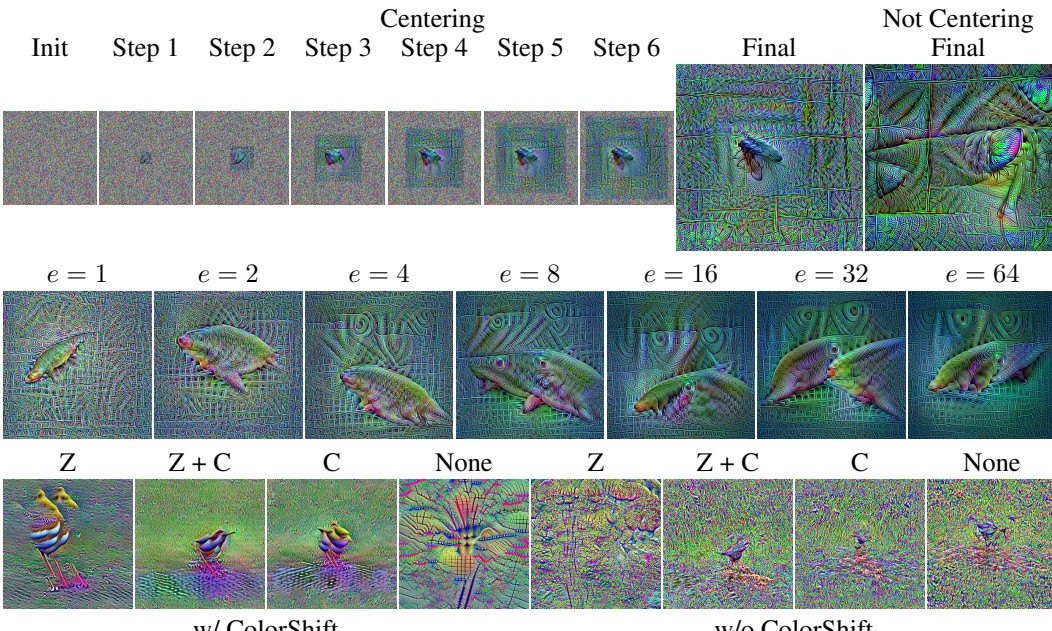

Figure 6: Ablation studies repeating the previous experiments for a *naturally*-trained ResNet-50. (top) Inversion for target class Fly with and without Centering. (center) Effect of ensembling size in the quality of inverted images for the Tench class. (bottom) The effect of various combinations of zoom, Centering, and ColorShift when inverting the Dipper class.

## 5 RESULTS

### 5.1 *PII* WORKS ON A RANGE OF ARCHITECTURES

We now present the results of applying Plug-In Inversion to different types of models. We once again emphasize that we use identical settings for the *PII* parameters in all cases.

Figure 7 depicts images produced by inverting the Volcano class for a variety of architectures, including examples of CNNs, ViTs, and MLPs. While the quality of images varies somewhat between networks, all of them include distinguishable and well-placed visual information. Many more examples are found in Figure 17 of the Appendix.

In Figure 8, we show images produced by *PII* from representatives of each main type of architecture for a few arbitrary classes. We note the distinct visual styles that appear in each row, which supports the perspective of model inversion as a tool for understanding what kind of information different networks learn during training.

### 5.2 COMBINING *PII* WITH DEEPINVERSION

We now consider the relative benefits of Plug-In Inversion and DeepInversion (Yin et al., 2020), a state-of-the-art class inversion method for CNNs. Figure 9 shows images from a few arbitrary classes produced by *PII* and DeepInversion (along with reference examples from the robust model demonstration in section 3). We additionally show images produced by DeepInversion using the output of *PII*, rather than random noise, as its initialization.

Using either initialization, DeepInversion clearly produces images with natural-looking colors and textures, which *PII* of course does not. However, DeepInversion alone results in some images that either do not clearly correspond to the target class or are semantically confusing. By comparison, *PII* again produces images with strong spatial and semantic qualities. Interestingly, these qualities appear to be largely retained when applying DeepInversion after *PII*, but with the color and texture improvements that image priors afford (Mahendran & Vedaldi, 2015).

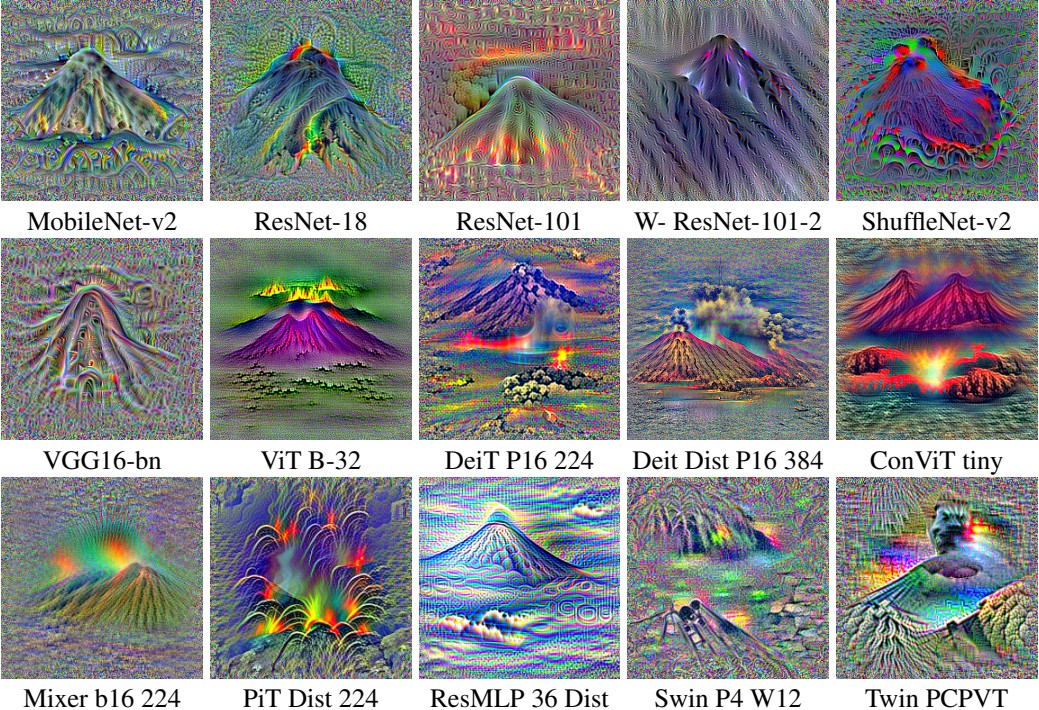

| MobileNet-v2 | ResNet-18 | ResNet-101 | W- ResNet-101-2 | ShuffleNet-v2 |
| VGG16-bn | ViT B-32 | DeiT P16 224 | Deit Dist P16 384 | ConViT tiny |
| Mixer b16 224 | PiT Dist 224 | ResMLP 36 Dist | Swin P4 W12 | Twin PCPVT |

Figure 7: Images inverted from the Volcano class for various Convolutional, Transformer, and MLP-based networks using *PII*. For more details about networks, refer to Appendix B.

## 6 CONCLUSION

We studied the effect of various augmentations on the quality of class-inverted images and introduced Plug-In Inversion, which uses these augmentations in tandem. We showed that this technique produces intelligible images from a wide range of well-studied architectures, as well as the recently introduced ViTs and MLPs, without a need for model specific hyper-parameter tuning. We believe that augmentation-based model inversion is a promising direction for future research in understanding computer vision models.

## 7 ETHICAL CONSIDERATIONS

We propose Plug-In Inversion as a class inversion technique for the purpose of understanding vision models. However, we note that prior work has considered the potential of model inversion to compromise the security of a model's training data (Fredrikson et al., 2015; Yin et al., 2020). These areas of progress and other data privacy concerns (Zhu & Han, 2020; Geiping et al., 2020) make clear the need for caution when sensitive data is used to train deep learning models.

## 8 REPRODUCIBILITY

All the models (including pre-trained weights) we consider in this work are publicly available from widely-used sources. Explicit details of model resources can be found in section B of the appendix. We also make the code used for all demonstrations and experiments in this work available at https://github.com/youranonymousefriend/plugininversion.

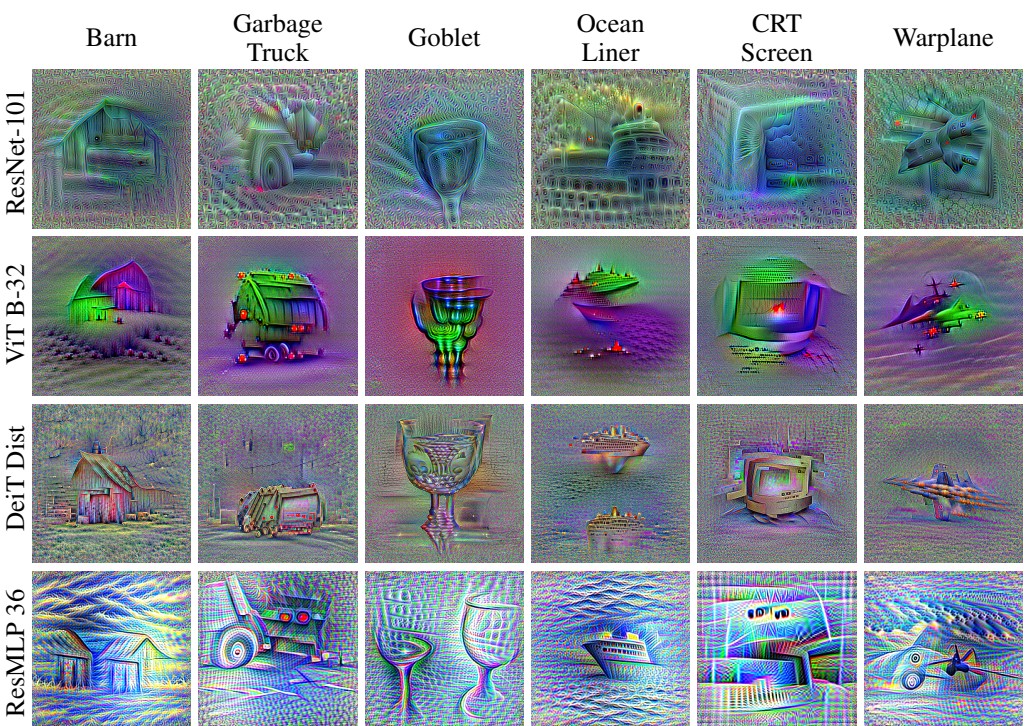

Figure 8: Inverting different model and class combinations for different classes using *PII*.

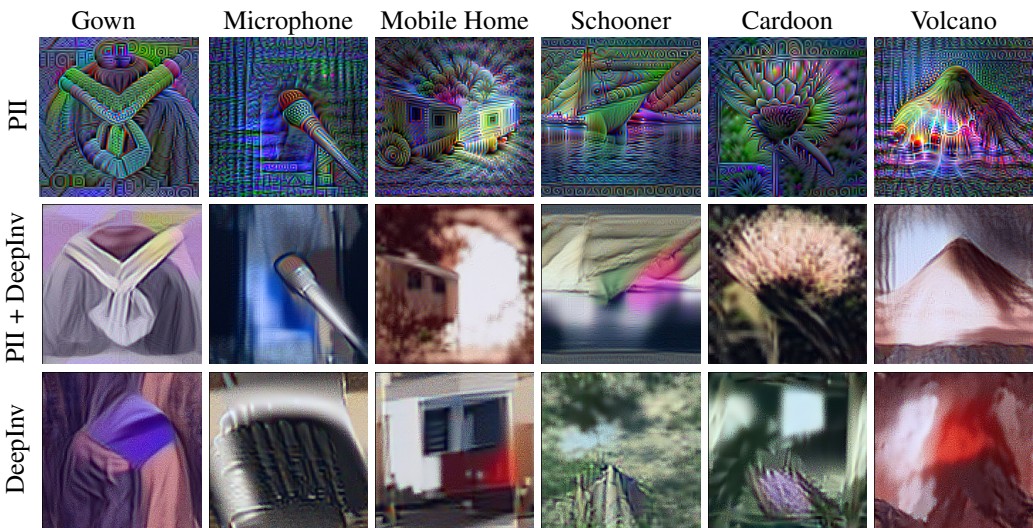

Figure 9: PII Inversion results for a naturally-trained ResNet-50. PII can be used as initialization for other methods such as DeepInversion.

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
