# OpenReview forum: "Plug-In Inversion: Model-Agnostic Inversion for Vision with Data Augmentations"
_ICLR.cc/2022/Conference — ICLR 2022 Submitted_

### Official Review · Reviewer_SYwW · 2021-10-28

**Correctness:** 3
**Technical Novelty And Significance:** 3
**Empirical Novelty And Significance:** 2
**Recommendation:** 5
**Confidence:** 4

**Main Review:**

# Strengths

* The authors aim for a model-agnostic and hyperparameter-robust method. This is an important direction.
* To verify the model-agnostic features, the authors tested the proposed method on various CNNs (MobileNet, ResNet, etc.), Vision Transformers (ViT, DeiT, etc.), and MLPs.

# Weaknesses

* According to Section 7, the purpose of this paper is to understand vision models. I agree with one of the goals of model inversion is so, but the paper lacks a discussion of how PII helps us to understand vision models, even though the authors conducted experiments on various image recognition models.
* The analysis is limited to qualitative ones. I would include quantitative results to compare PII with other methods. For example, DeepInversion [Yin+20] reported classification accuracy and the Inception score (IS). Santurkar+19 also shows IS.
* The authors claimed that DeepInversion needs statistics $\hat{\mu}_k, \hat{\sigma}^2_k$ ported from batch normalization (BN) and thus is not applicable to models sans BN, such as ViT. However, I think that $\hat{\mu}_k, \hat{\sigma}^2_k$ are statistics of training data and can be obtained even for models sans BN only by passing training data once and storing the activations.
* PII is only verified on the ImageNet dataset. I recommend the authors to use CIFAR-10 as Yin+20.

# Comments

* References are not well organized and written in a consistent format. For example, Santurkar+19 is accepted at NeurIPS 2019, but it is not written so.
* An image is denoted as $x$ in p. 4, but $\mathbf{x}$ in P5. The notation should be consistent, otherwise, it is difficult to follow.
* Why a total-variation loss is used to visualize Figure 4, while, if I understand correctly, PII does not use the loss term?
* Without a quantitative measure, how the hyperparameters of PII can be tuned in Section 4?
* Section 3.3 says $e=8$ results in images of acceptable quality, while in the main experiment, $e$ is set to 32. If it is intended, it would be justified.
* What are "random augmentations" in p. 6 Section 4?

**Summary Of The Paper:**

This paper proposes Plug-In Inversion (PII), which can invert a trained image classifier so that it generates a class-conditional image.
In addition to CNNs that previous work has considered, PII can handle Vision Transformers and Vision MLPs as well.
PII starts generating from the center of a random initialization with lower resolution. Then, the method gradually broadens the generating area (centering) and increases the resolution (zooming). During generation, colors are randomly shifted (ColorShift) and the averaged cross-entropy to $e$ color-shifted images are used.



**Summary Of The Review:**

The model aims for a model-agnostic and hyperparameter-robust method to investigate vision models by model inversion. I highly appreciate this goal. Meanwhile, the current manuscript has several weaknesses. Therefore, I recommend this paper be marginally below the threshold.

---

> ### Author Response · Authors · 2021-11-22
> **Thank you for your feedback, reviewer SYwW!**
>
> Thank you for your questions and comments! Please see our responses below, as well as our revised submission, particularly the new sections E-H in the supplementary material.
>
> Per your suggestion, we have added quantitative analyses (Section H), specifically classification score of a variety of models, as well as Inception score, on images inverted from a ViT model and from a ResMLP model using PII and DeepDream (Mordvintsen et al. 2015; Yin et al. 2020). We find that PII uniformly outperforms DeepDream when applied to the ViT and achieves higher Inception scores and comparable classification accuracy when applied to the ResMLP.
>
> We have additionally updated the submission with results for CIFAR-10 and CIFAR-100 (see Section F), produced using identical settings to those we used for ImageNet. We will continue collecting results on additional models to include in a final version.
>
> You make a good point regarding BN statistics - with access to training data, one could produce these statistics for any model and use them for the feature regularizer in DeepInversion. However, DeepInversion is proposed as a data-free method (as is PII), and so it seems appropriate not to assume such access.
>
> TV regularization was used in Figure 4 to illustrate that inversion using ColorShift is not sensitive to this term, justifying its removal from the objective.
>
> The PII hyperparameters $\alpha$ and $\beta$ were tuned using qualitative experiments like the one shown in Figure 10 of the appendix, and our main results show that the chosen values do indeed transfer well to other models.
>
> Regarding the ensemble size $e$, we mean to convey that smaller values than 32 still yield positive results, but that we opt for a larger size when GPU constraints allow. The statement about ‘different random augmentations’ means that the augmentations that involve randomness (jitter and ColorShift) are re-drawn at each gradient step. We have updated the current draft to clarify these points as well as to fix notational issues you and other reviewers mentioned.

---

> > ### Comment · Reviewer_SYwW · 2021-11-24
> > **Thank you for your reply**
> >
> > Thank you for your reply.
> >
> > * CIFAR-10 and CIFAR-100 results: The results seem convincing that the PII can invert classes well.
> > * BN statistics of DeepInversion, TV term in Fig.4, hyperparameters: Thank you for clarifying my concerns.
> >
> > I still have the following comments, which I hope help the authors to improve the manuscript.
> >
> > * As of the quantitative results, I do not fully understand if classification accuracy is the metric to be used.
> > * I think that some ablations (e.g., Fig. 5) would be moved to the appendices, while more important parts (qualitative results?) would be moved to the main text.
> > * As written in the original review, the purpose of this research would still not be clear enough. If it's clear, the abovementioned comments would be resolved.

---

> > > ### Author Response · Authors · 2021-11-25
> > > **Thank you for additional feedback!**
> > >
> > > Thank you for taking the time to look over our response and the updated draft! We appreciate the additional feedback.
> > >
> > > Regarding the purpose of this research: we frame class inversion in this paper in the context of understanding what a model has learned during training. This perspective is consistent with other work in this area. For example, data-free methods such as ours can give insight into the nature of images in the (possibly private) dataset on which a model was trained (Frederikson et al. 2015; Yin et al. 2020). Also, producing multiple images for the same class helps to see invariances of a model’s representation (Mahendran & Vedaldi 2015). We propose PII as a tool to accomplish this, and one which is more broadly applicable than prior techniques, expanding the range of models which can be explored. Additionally, we think an interesting future direction is to compare directly between what visual information tends to be captured by different architectures. This would require a uniform method like PII (which is applied identically to any model) to ensure observed differences may be attributed to architecture rather than choice of hyperparameters for inversion. Although we can no longer upload revisions to OpenReview, we have updated the introduction, background, and conclusion to make this discussion more prominent.
> > >
> > > Similarly, we have moved some of the quantitative results into the main paper and relegated some of the less crucial figures (5 and 6) to the appendix.
> > >
> > > Classification score has previously been used as a quantitative measure of inverted images in Yin et al. (2020). We feel that this measure is a way to confirm that the images capture visual information that is recognizable as corresponding to a particular class, rather than possibly overfitting to the peculiarities of a particular model’s representation. We additionally report Inception score, which has been extensively used as a measure of generated image quality, including for inversion (Santurkar et al. 2019; Yin et al. 2020).

---

### Official Review · Reviewer_vGig · 2021-10-31

**Correctness:** 4
**Technical Novelty And Significance:** 2
**Empirical Novelty And Significance:** 3
**Recommendation:** 6
**Confidence:** 3

**Main Review:**

Strengths of the paper:
- The proposed method works for class inversion for any architecture, including Vision Transformers and MLPs.
- The proposed method is robust across architectures with little-to-no hyperparameter tuning

Weak points of the paper:
- The specific method proposed is not clearly presented at a mathematical level (it is described only in words)
- The only baseline comparison I saw was relative to one method (DeepInversion).

Requested response from authors that will affect my recommendation:

The authors should justify why they believe their baseline comparisons are sufficient (or comment on what additional baselines they would present in a camera ready).

The paper does not comment on methods for adversarial examples, such as adversarial patches, either as related work or as baselines.  In the rebuttal, please justify why these comparisons are not present, or comment about them.

The authors should clarify what exactly the PII algorithm is at a mathematical level (write out the GD iteration) so that I am clear I am not misunderstanding the model.

Additional feedback with the aim to improve the paper:

If my reading is correct, the method treats the classifier as a black box, which is why it works for any architecture.  If this is true, then the paper would be much improved by clearly stating it treats the classifier as a black box.  In its current form, it comes across as being restricted to neural networks, which I do not think is the case.

Typos:
The minimization problem on page 2 should say argmin_x L(f(x), y).  It has an incorrect parenthesis.

There are several places where the document says "cite XYZ".  Replace these with the citations.

Figure 2 caption: "An image a different stages" has a grammatical issue.


**Summary Of The Paper:**

In this paper, the authors introduce a novel method for the class inversion problem: given a trained classifier and a specific class/label, generate an example image of that class.  This method is called Plug-In Inversion (PII) and it works as follows: a gradient descent is performed to find the image x such that the output of the classifier is close to the target label under a suitable loss function.  In PII, each successive gradient update is computed using a data-augmented version of the current iterate of the image x.  The authors introduce new data augmentations and combinations thereof.  The benefits of this inversion method is that it works for any classification architecture (which is treated only as a black box), and the same hyper parameter values (e.g. amount of color shift) work across a range of significantly different classifier architectures.  The authors present the output of PII across 12 different networks (including Convolutional, Transformer, and MLP nets), and they demonstrate several examples where their method is more interpretable than DeepInversion, a state-of-the-art method for class inversion.


**Summary Of The Review:**

Clearly state your recommendation with 1-2 reasons

I rate this paper as a weak accept for the following reasons.

The reason to accept is: They provide a class inversion method that produces interpretable visualizations for recent architectures (Vision Transformers + MLPs) that existing methods do not work for.  This could accelerate the development of better classifiers by helping researchers interpret and fix their behavior.

The reason this isn't rated higher is: The baseline comparison was inadequate.  The authors should compare their visualization method on Transformers/MLPs to the best existing approach that exists before this work.  Perhaps that approach is a plain inversion method.  Perhaps there are methods that can be borrowed from the methods of targeted adversarial examples.

---

> ### Author Response · Authors · 2021-11-22
> **Thank you for your feedback, reviewer vGig!**
>
> Thank you for your questions and comments! Please see our responses below, as well as our revised submission, particularly the new sections E-H in the supplementary material.
>
> We center our discussion around DeepInversion because it is a strong and recent method for data-free class inversion of naturally trained models. Additionally, it incorporates techniques from multiple previous works (Frederikson et al. 2015; Mahendran & Vedaldi 2015; Mordvintsen et al. 2015). Other methods we reference are either not data-free (Simonyan et al. 2015) or are specific to robust models (Santurkar et al. 2019). However, we agree it is helpful to directly compare to a baseline for non-CNN models, so we have added images generated by DeepDream (i.e., DeepInversion minus the feature regularizer) to parallel Tables 7 and 8 (see section G), as well as quantitative comparisons between that method and ours (section H).
>
> We have now included pseudocode for the optimization procedure (Section E), thank you for this suggestion!
>
> Could you please clarify your question regarding methods for adversarial examples? It is unclear to us how this would be incorporated in the data-free setting.
>
> While we intend this method to be model-agnostic, it is not a black-box method because we need to backpropagate through the model to get the gradient with respect to the input.
>
> Finally, thank you for pointing out notational errors; we have fixed these in our updated draft.

---

### Official Review · Reviewer_MNJX · 2021-11-01

**Correctness:** 3
**Technical Novelty And Significance:** 3
**Empirical Novelty And Significance:** 2
**Recommendation:** 6
**Confidence:** 4

**Main Review:**

This paper proposes a technique called Plug-In Inversion, which is to be used as-is, irrespective of the underlying model which is to be inverted. This method works by applying a particular sequence of augmentations to the image, and then performing inversion by minimizing the loss of this image with respect to a single class.

Strengths:
- The motivation behind the paper is clearly presented - the goal is to reduce the need for extensive hyperparameter optimization of an inversion system. This is an interesting motivation, given that extensive tuning of regularizers is often difficult to perform.
- The paper is in general clear and easy to follow. The main ideas are presented in a concise manner and formulate a solid story for the paper.
- The authors demonstrate the effects of each different augmentation used in their inversion system, both in the main paper and the appendix. This is particularly useful both for understanding how Plug-In Inversion works and how each part of the procedure affects the resulting image (namely, how the zoom + centering procedure operates, as well as the effect of the ColorShift augmentation).
- The proposed method is evaluated on a great variety of models, and produces intelligible images for the classes presented (in the sense that they conform to the high-level notion related to the class).

Weaknesses:
- The proposed Plug-In inversion method is introduced very late in the paper (at the end of Section 3.4) even though it consists only of combining the augmentation and search space restriction techniques provided in the previous sections (3.1 - 3.3). It would have been much clearer for the reader if this fact was fully described earlier in Section 3 (for example, by moving Section 3.4 earlier in the paper), instead of delaying the presentation of the full method.
- My greatest concern is the fact that I am not sure whether the claim of the authors that the method can be applied to multiple networks without tuning is fully supported by the experimental results. More specifically, in Figures 7 and 8 the method is applied in a variety of networks, resulting in images which, while intelligible, are of varying quality. As such, to fully support the argument of the lack of need of extensive hyperparameter tuning, one would also need to show that using the same regularizer (for example, TV, which can be applied to any model) with the same hyperparameter leads to more extensive degradation, when applied to different models. I believe that a small example to demonstrate this motivation would greatly improve the paper.
- The above point is made even more unclear by the fact that one of the models considered by the authors does use a TV regularizer (which leads to drastic improvement in image quality).
- The ColorShift augmentation proposed by the authors is, unless I am mistaken, a variant of color jittering (in the sense that the adjustment is made directly to the pixels of the image, rather than on hue, saturation, contrast etc.). Given that applying some form of adjustment to the color of the image is a form of data augmentation which has been used in prior work  (Krizhevsky et al., 2012, section 4.1), I am not sure if this data augmentation is as novel as the authors claim (although I do appreciate the qualitative analysis of its effects in the context of inversion).

Questions:
-	The hyperparameters for ColorShift are fixed to $a=b=1$, and if I understand correctly this comes from a qualitative analysis of the results in a few simple experiments. Is this correct?

Minor comments/typos:
-	As a minor comment related to the above, I believe the authors should indicate in the captions of the figures which model the respective images come from.
-	There is a space missing in the second-to-last line of page 6.
-	The caption in Figure 10 in the appendix is incomplete.

References:
Krizhevsky, A., Sutskever, I., & Hinton, G. E. (2012). ImageNet Classification with Deep Convolutional Neural Networks. Advances in Neural Information Processing Systems, 25, 1097-1105.


**Summary Of The Paper:**

This paper proposes a method to perform class inversion on image data, called Plug-In Inversion. This method consists of a sequence of augmentations on image data and is designed to be applicable to a variety of architectures. This method is evaluated on ImageNet trained models and is compared with other techniques used for the same goal.

**Summary Of The Review:**

Overall, this work has an interesting motivation (alleviating the need for extensive tuning of regularizers for class inversion), but I am not certain if the benefits of Plug-In Inversion, as described by the authors, are fully supported by the experimental evidence provided. As such, given also the fact that the individual components of the method do not seem novel by themselves, I marginally lean towards rejecting this paper, pending extra discussion during the rebuttal process.

**Update after rebuttal**: See response to the authors' comments below.

---

> ### Author Response · Authors · 2021-11-22
> **Thank you for your feedback, reviewer MNJX!**
>
> Thank you for your questions and comments! Please see our responses below, as well as our revised submission, particularly the new sections E-H in the supplementary material.
>
> We have added several results which we hope address your central concern. First, we present results on CIFAR-10 and CIFAR-100 (Section F) which use identical settings to those we used for ImageNet; compare to DeepInversion (Yin et al., 2020), which tunes a specific set of regularization weights for each dataset (some of which differ by several orders of magnitude). We have also added images generated by DeepDream (i.e., DeepInversion minus the feature regularizer) to parallel Tables 7 and 8 (see section G), which shows that using this regularization-based approach with fixed weights does not transfer as well across models as PII, and further validate this with a quantitative evaluation in section H. We also wish to clarify that the improvement you attribute to TV regularization is actually due to that model being robustly-trained; note that Figure 4 shows that when using ColorShift, TV regularization has no observable effect.
>
> Yes, you are correct - the PII hyperparameters $\alpha$ and $\beta$ were tuned using qualitative experiments like the one shown in Figure 10 of the appendix, and our main results show that the chosen values do indeed transfer well to other models.
>
> We agree that, broadly speaking, pixel-level color adjustment is already in use as an augmentation method. We feel that the novelty of our method is in deliberately distorting the color from the distribution one would expect for natural images in order to improve inversion (rather than simulating small differences in lighting conditions, etc.).
>
> Your feedback regarding how the overall method is introduced is helpful. We have added pseudocode for the optimization procedure (see Section E), and hope that this will be helpful in making the presentation more clear.
>
> We also thank you for other notes regarding notation and writing.  We have updated our current draft to reflect your comments, and we will continue to improve our draft for the camera-ready version.

---

> > ### Comment · Reviewer_MNJX · 2021-11-25
> > **Raising my score slightly.**
> >
> > Thank you very much for your responses.
> >
> > I can see from the new experiments on CIFAR-10 and CIFAR-100 datasets that PII can be applied to a general setting. I also appreciate the clarification with respect to the robust model, and the actual role of the TV regularizer. I have raised my score slightly - the quality of the generated images is more consistent across the models for the other datasets, which is in support of the main claim of the paper.
> >
> > I still have some reservations, given that the motivation and the conclusions obtained would have been clearer if the paper also included an analysis of the effect of using a regularizer on the non-robust model. Even if the improvements in image quality are due to the robust training instead of the regularizer, demonstrating some examples of the use of the regularizer (by itself, on the non-robust model) would have helped clarify the message of the paper.

---

> > > ### Author Response · Authors · 2021-11-25
> > > **Thank you for additional feedback!**
> > >
> > > Thank you for taking the time to look over our response and the updated draft! We appreciate the additional feedback.
> > >
> > > Regarding the effect of the regularizer: we wish to clarify that the baselines of DeepDream/DeepInversion that we now compare to in sections G and H are regularizer-based methods. In particular, both of them include a TV regularization term. We thus believe the images in Figures 26-27 and the quantitative comparison in Figures 28-29 (which are produced from non-robust models) are examples of what you are suggesting. We are also now producing analogous figures to fig. 4 and fig. 11 (replacing the robust model with a non-robust one), which should further isolate the effect of the regularizer. Thank you for this suggestion!

---

### Official Review · Reviewer_dsJC · 2021-11-03

**Correctness:** 2
**Technical Novelty And Significance:** 2
**Empirical Novelty And Significance:** 1
**Recommendation:** 3
**Confidence:** 4

**Main Review:**

# Highlights:
The problem addressed in the paper is well written and well described in the Introduction and Background section.

# Related work:
There should be a separate Related work section citing the work that has been done in this area previously with a thorough literature review.


# Methods:
1.	The paper mentioned several drawbacks in the Introduction section of the existing methods - the authors should put any of such examples like the generated images being highly sensitive to the weights assigned to the regularized terms, etc, and compare that with the images generated by PII to show how PII is working better.
2.	The authors should have provided some theoretical justifications of their work -  why are the new sets of augmentation working better than its predecessor?

# Experiments:

1.	One of the main issues in this paper is the applicability of the solution of the problem is not discussed in the paper. For example, there are three major applications where Deep Inversion(Yin et al., 2020) can be used - (a) pruning, (b) knowledge transfer, and (c) continual (class incremental) learning. If the authors set Deep Inversion as the baseline, they should have formulated the experiments for these three applications and compared PII with all architectures like VIT and MLP and others against Deep Inversion. Also if there are any other applications of the solution, that should have been clearly mentioned in the paper.

2.	The authors experimented only using the Imagenet dataset. They should have done against other datasets like CIFAR 10 - the way Deep Inversion was doing.

3.	To prove such a simple method is working better than the previous models, the authors should have either used or introduced some metric that gives a quantitative comparison between all the inversion techniques. In this paper, the authors compared PII with different settings and against Deep Inversion qualitatively. For example, in Deep Inversion, the authors did a wide range of quantitative comparisons among various metrics like inception score, knowledge transfer results, continual learning results (table 1-7 Deep Inversion (Yin et al., 2020))

# Minor:
1.	Section 2.3, 2nd paragraph where Vision transformers are discussed - in line #2, there is a mistake \cite{Vaswani}
2.	Same for the last line in the same paragraph \cite{CoaT and CaiT}


**Summary Of The Paper:**

This paper attempts to solve the problem of image inversion by introducing three augmentation-based techniques. It specifically tries to solve the problem of class inversion, which generates an interpretable image from a neural network by sending the pre-initialized image to the network. Then optimization is wrt the input image instead of the weights of the network. It introduced new sets of augmentation-based techniques like centering, zooming, color shift augmentation, ensembling that have not been tried before, and finally, they combined them with VIT and MLP based vision models. They validated all the augmentations using Vision transformer and MLP based vision models and compared them against the exiting method Deep Inversion(Yin et al., 2020). The paper is well written but lack quantitative evaluation which is a very significant shortcoming.

**Summary Of The Review:**

* Since the baseline for this paper is Deep Inversion lack and there are many quantitative experiments in that paper, the lack of quantitative results in this paper is not justifiable.

---

> ### Author Response · Authors · 2021-11-22
> **Thank you for your feedback, reviewer dsJC!**
>
> Thank you for your questions and comments! Please see our responses below, as well as our revised submission, particularly the new sections E-H in the supplementary material.
>
> Prompted by your comments, we have now added comparisons to a regularizer-based method which shows that the same hyperparameters optimized for CNN models do not yield good results for ViT and MLP models (see Section G) to help illustrate these drawbacks. We also refer to Figure 4, which shows that inversion is insensitive to the TV regularization weight when using our ColorShift augmentation, but sensitive without it.
>
> Along similar lines, we now present CIFAR-10 and CIFAR-100 results (see Section F) that are produced using identical settings to those used for ImageNet; compare to Yin et al. (2020) where hyperparameters are tuned (to significantly different values) for each dataset.
>
> While we discuss DeepInversion in the paper and consider it to be the state-of-the-art method for inverting CNNs, we emphasize that it cannot be applied to ViTs and MLPs; as such, it is not possible to directly compare our method to theirs on the vast majority of models we consider. We also do not propose PII as a method for the specific applications you mention; we focus on class inversion in the context of visualizing what models have learned during training. Nonetheless, we agree that it is worth performing quantitative evaluations, and we now report classification and Inception score results for PII vs. DeepDream (i.e., DeepInversion minus the feature regularizer) applied to a ViT and an MLP in Section H, which we find to clearly favor PII.
>
> Finally, thank you for pointing out the incomplete citations; we have corrected them in the updated submission.

---

> ### Author Response · Authors · 2021-11-26
> **Checking back**
>
> Hi reviewer dsJC,
>
> Do you have any additional feedback we can address during the discussion period?

---

### Author Response · Authors · 2021-11-29
**Summary of author response to reviews**

We wish to express our thanks to all the reviewers for their extremely helpful feedback, which we believe has helped us to substantially improve this work. Due to the extent of these improvements, we felt it would be helpful to give a clear summary of the most important additions and changes that were made.

**Additional datasets:**
Two of the reviewers indicated that only validating Plug-In Inversion on ImageNet was insufficient, and suggested that results for CIFAR-10 should also be included. We collected results from 4 ViT models for each of CIFAR-10 and CIFAR-100, which are pictured in Figures 24-25. We emphasize that these were produced using identical settings to the ImageNet results, in contrast to methods which tune hyperparameters to different datasets, which we feel greatly strengthens our central claim.

**Quantitative results:**
Multiple reviewers also pointed out that quantitative evaluations would help to give a more objective comparison to prior methods. We now report Inception score and cross-model classification accuracy on images produced by PII and by DeepDream from a ViT and from an MLP (figures 28-29). These evaluations clearly favor PII over DeepDream, the best comparable inversion method we are aware of.

**More thorough baseline comparisons:**
We also have added additional images produced by DeepDream and DeepInversion (shown in figures 26-27) to directly compare to those from PII in figures 7-8. These images illustrate that while PII gives consistent results across models (and datasets, as discussed above) with the same settings, the alternative regularizer-based approaches require hyperparameter tuning.

**Formal description of the algorithm:**
At the suggestion of reviewer vGig, we have included pseudocode for the algorithm, which we hope will help clarify how PII is formulated mathematically.

---

### Comment · Reviewer_dsJC · 2021-11-30
**after rebuttal**

# Highlights:
- The authors have included the results of CIFAR 10 and CIFAR 100 datasets
- The authors have added comparisons to regularized based method using CNN and compared against MLP and VIT models
- The authors added IS for PII

# Comments:
1. The results for CIFAR datasets show improvements compared to the older methods.
2. The comparisons with regularized based method showed improvements over CNN models.
3. For, IS they only reported the score, they should put a table with details from all the previous models.
4. For ResMLP (Fig 29), shows bad results or similar results for PII over DeepDream.

I am still not convinced by the aim of this paper. Like in the original review, DeepInversion has some applications. If this paper does not want to address those, is there any other purpose that the authors want to address? Also they are viewing the problem of class inversion in the context of what the models are learning during training. So if the authors are focusing on the explainability or interpretability part, they should discuss why their method is better than the other methods in terms of explanability; they should do a detailed comparitive study and i beleive there is room for including the both qualitative and quantitative results while doing the comparison. In summary the purpose and the applicability of this research is still ambiguous.

---

> ### Author Response · Authors · 2021-11-30
> **Thank you for additional feedback!**
>
> Thank you for taking the time to look over our response and the updated draft! We appreciate the additional feedback.
>
> We agree with your statement that we are viewing class inversion in the context of what models learn. This perspective is consistent with other work in this area. For example, data-free methods such as ours can give insight into the nature of images in the (possibly private) dataset on which a model was trained (Frederikson et al. 2015; Yin et al. 2020). Also, producing multiple images for the same class helps to see invariances of a model’s representation (Mahendran & Vedaldi 2015). We propose PII as a tool to accomplish this, and one which is more broadly applicable than prior techniques, expanding the range of models which can be explored. Additionally, we think an interesting future direction is to compare directly between what visual information tends to be captured by different architectures. This would require a uniform method like PII (which is applied identically to any model and dataset) to ensure observed differences may be attributed to architecture rather than choice of hyperparameters for inversion.
>
> Much of the prior work in class inversion focuses primarily on the quality of inverted images to support that the proposed methods achieve their stated interpretability/explainability goals. In line with this, the qualitative and quantitative evaluations in our work, such as those you mention in your comment, demonstrate that for a fixed set of hyperparameters, our method produces images of equal or better quality than the current state-of-the-art when applied to non-CNN architectures (and this improvement holds across datasets as well, which you also note). Since image quality is paramount to the above-mentioned objectives, we feel this is a meaningful advancement.
>
> Although we can no longer upload revisions to OpenReview, we have updated the introduction, background, and conclusion to make this discussion more prominent.

---

### Decision · Program_Chairs · 2022-01-20

**Decision:**

Reject

**Comment:**

This paper presents a new method for solving the problem of inverting image classifier models. The authors introduce three new augmentation-based techniques to do this. The techniques are validated using Vision Transformer and MLP models and compared against previous methods. The reviewers appreciate the problem that the paper aims to solve. However, the reviewers are not satisfied with the presentation and evaluation of the proposed approach. The main contribution of the paper is not presented clearly enough, according to the reviewers, and it remains unclear to them what aspect of model inversion the authors most want to improve on, and whether their proposed technique indeed achieves such an improvement. In their response, the authors do provide Inception scores that show that their inversion method improves the perceptual quality of generated images compared to previous approaches. The reviewers acknowledge the author response, but indicate that it does not fully resolve their concerns. I recommend that the authors update their paper to more clearly present their main contributions and conclusions, and to provide a more thorough comparison against previous methods, before submitting to another conference.